# Spatial and Temporal Evolution Analysis of Industrial Green Technology Innovation Efficiency in the Yangtze River Economic Belt

**DOI:** 10.3390/ijerph19116361

**Published:** 2022-05-24

**Authors:** Mengchao Yao, Jinjun Duan, Qingsong Wang

**Affiliations:** DongWu Business School, Soochow University, Suzhou 215006, China; fredbs@126.com (J.D.); sudakjc@163.com (Q.W.)

**Keywords:** industrial green technology innovation, super-SBM model, GML index, kernel density estimation, spatial Markov chain

## Abstract

As a fusion point of innovation-driven green development, green technology innovation has become an essential engine for green transformation and high-quality economic development of the Yangtze River Economic Belt. Based on the panel data of 110 cities in the Yangtze River Economic Belt from 2006 to 2020, this paper uses the super-SBM model to measure the efficiency of industrial green technology innovation. Then, the Dagum Gini coefficient and its subgroup decomposition method, kernel density estimation, and the spatial Markov chain will discuss the convergence characteristics and dynamic evolution law of industrial green technology innovation efficiency in the Yangtze River Economic Belt. The results indicate several key points. (1) On the whole, the industrial green innovation efficiency of the Yangtze River Economic Belt shows a trend of the “N” type, which increases slowly at first and then decreases and then increases, and shows a non-equilibrium feature of “east high and west low” in space. (2) The average GML index of industrial green technology innovation efficiency in the Yangtze River Economic Belt is greater than 1, and technological progress is the main driving force in promoting efficiency growth. (3) There are spatial and temporal differences in industrial green technological innovation efficiency in the Yangtze River Economic Belt. Interregional differences and hypervariable density are the primary sources of overall differences. (4) During the study period, the absolute difference in industrial green technology innovation efficiency among regions showed a trend of “expansion-reduction-expansion”, and the innovation efficiency gradually converged to a single equilibrium point. (5) The industrial green technology innovation efficiency transfer in the Yangtze River Economic Belt shows a specific spatial dependence. Accordingly, policy suggestions are put forward to further improve industrial green technological innovation in the Yangtze River Economic Belt.

## 1. Introduction

With the increasingly prominent contradiction between economic development and the resource environment, green development has become an inevitable choice for China’s economic sustainable development. Green technology innovation is the core driving force of green growth. As the leading force of the national economy, industrial enterprises are the primary resource consumers and pollution emitters, facing tremendous pressure and challenges of green transformation [1]. Thus, green technology innovation in China’s industrial green development has become increasingly prominent, which has become an essential means for industrial enterprises to deal with the crisis of resource consumption, solve the problem of environmental pollution, and realize green transformation and development. The Yangtze River Economic Belt, which has one of the three national strategies, is one of the regions with the most vital comprehensive strength and the most significant strategic support in China [2]. Industry occupies a pivotal position, especially the manufacturing industry, which has become the lifeline of the Yangtze River Economic Belt and national economic development. However, the traditional sector still occupies a high proportion for a long time. Its economic growth model is still at the cost of large resources and environmental damage, resulting in the contradiction between economic growth, resources, and environmental carrying capacity. The ecological environment problems in various regions are increasingly prominent [3]. For example, soil erosion in the upper reaches, wetland degradation in the middle spaces, and excessive development in the lower reaches have severely restricted high-quality economic growth [4]. This study is significant because, in the face of severe ecological and environmental problems, it is necessary to measure the level of green industrial innovation in the Yangtze River Economic Belt and analyze the spatial evolution characteristics, convergence or divergence, and the spatial dynamic evolution of industrial green innovation efficiency. It is of great strategic significance to provide a reference for the formulation and improvement of environmental policy tools to effectively improve regional industrial green innovation and realize the “win-win” of economic development and environmental protection.

As a fundamental means of coordinating economic development and environmental protection, green technology innovation is developed based on traditional and green innovations. Green technology innovation, also known as ecological technology innovation, is a form of technological innovation. It refers to the general term of technologies, processes, and products that follow the basic principles of ecological environment and ecological economy, save resources, reduce energy consumption, and eliminate environmental damage as far as possible to ultimately minimize negative environmental effects [5]. Green technology innovation efficiency, environmental innovation, sustainable development, and ecological innovation have the same meaning [6]. The purpose is to solve the contradiction between environmental constraints and economic growth. In a narrow sense, the efficiency of green technological innovation is different from that of innovation in the general sense, and it highlights the “technology” characteristics [7]. In terms of evaluation methods, it is generally measured based on technical efficiency and measured by the input–output approach.

Existing research on green technology innovation mainly focuses on the following aspects. First, as shown by evaluating the efficiency of green technology innovation, the existing studies are mainly based on the level of enterprises or industries. Taking heavily polluting enterprises as the research object [8], Hu et al. [9] discussed the impact of green credit policy (GCP) on green technology innovation. Xu et al. [10] measured the green technology innovation efficiency and green total factor productivity of the advanced manufacturing industry in the Yangtze River Delta by constructing a two-stage evaluation index system and analyzing its spatial differentiation characteristics. Li et al. [11] built a network SBM-DEA model considering undesired output based on the two-stage perspective of iron pre-process and post-iron process, measured the green technology efficiency of Chinese iron and steel enterprises from 2009 to 2013, and discussed the impact of environmental regulation intensity on green technology efficiency. To improve the chemical industry’s green innovation and development efficiency, Zeng et al. [12] used the SBM-undesirable model to measure the green innovation efficiency of the chemical industry in the Yangtze River Economic Belt and used geographic detectors to analyze the driving factors of its spatial differentiation. Second, in measuring green technology efficiency, the research mainly focuses on the parameter-based stochastic frontier model and the non-parameter-based data envelopment analysis method [13], which are more suitable for multi-input, single-output research [14,15]. Harvie et al. [16] used SFA to evaluate the technological innovation efficiency of small- and medium-sized enterprises in Thailand and concluded that labor-intensive enterprises with low technical efficiency dominated Thailand. To examine the role of green technology innovation in economic growth and environmental issues, Hu et al. [17] used a modified stochastic frontier model (SFA) based on Translog and output distance functions to measure China’s green technology innovation efficiency. Wang et al. [18] made improvements based on the original SFA model and combined the characteristics of the projection pursuit model to process high-dimensional data, and constructed a more effective calculation model to analyze the green innovation technology innovation of 30 provinces and cities in China from 2005 to 2011. Third, with growing research and increased attention on environmental problems in academia, it is often impossible to obtain a more comprehensive level of green technology innovation using a single output index. The data envelopment analysis model based on a non-parameter can effectively calculate multi-input and multi-output variables. They set specific functions to create a mainstream industrial green innovation efficiency calculation method [19,20,21]. For example, Wang [22] used the DEA-RAM model to study the efficiency of green technology innovation in 29 industries of China’s manufacturing industry from 2005 to 2012 and concluded that the innovation of China’s manufacturing industry is changing to green innovation and the contribution to green growth is increasing. CL A [23] combined with the global Malmquist–Luenberger (GML) index of the data envelopment analysis (DEA-SBM) model was used to measure the green technological innovation efficiency of 30 provinces in China from 2003 to 2017. Tan et al. [24] measured the green technology innovation efficiency of core cities in the Yangtze River Delta region from 2010 to 2017 by constructing a super-efficiency SBM-DEA model, including undesired outputs, and studying its spatiotemporal evolution pattern driving factors. Zuo et al. [25] measured and analyzed the technological innovation efficiency of my country’s inter-provincial industrial rate from 2008 to 2017 by constructing a two-stage DEA model based on the heterogeneity of interval technology to make scientific improvements, the level of technological innovation provides theoretical support. 

Scholars have carried out multi-dimensional and multi-perspective studies, discussed the evaluation and calculation of industrial green technology innovation efficiency, and achieved specific results which laid a theoretical and methodological foundation for this study. However, there are still the following issues worthy of discussion. (1) On the research scale. The existing research is mainly concentrated on the national and provincial levels, and less based on the urban level. (2) Green innovation has a typical characteristic of “dual externality” and its size is closely related to regional economic development level, industrial structure, resource endowment, R&D investment, etc. However, existing research still lacks any analysis of regional heterogeneity. (3) In the research method, the traditional DEA model does not consider the “relaxation” factor and cannot reasonably solve the efficiency evaluation problem under the condition of the existence of undesirable outputs [26]. (4) In the industrial green innovation efficiency analysis process, more emphasis is placed on static evaluation, and the research on the dynamic evolution trend is relatively lacking. This paper takes 110 prefecture-level cities in the Yangtze River Economic Belt as the research object to solve the above problems. It uses the super-SBM model and the Malmquist–Luenberger index model to analyze the evolution of industrial green innovation efficiency from regional heterogeneity. It then uses Dagum Gini coefficients to reveal the spatial and temporal differences and sources of industrial green technology innovation efficiency. Furthermore, the spatial factors are included in the research framework, and the spatial and temporal evolution of green innovation efficiency is analyzed by kernel density and the Markov chain method. The aim was to provide a targeted decision-making basis for improving the efficiency of industrial green technology innovation in different Yangtze River Economic Belt cities and promote the green and high-quality development of the Yangtze River Economic Belt economy.

This paper is organized as follows. The second part discusses the research design. The super-SBM model, the GML index decomposition method, kernel density estimation, and spatial Markov chain are introduced. The third part offers an analysis of the empirical results. It mainly includes the size and spatial distribution of the industrial green technological innovation level in the Yangtze River Economic Belt, the temporal and spatial evolution trend of efficiency, and the spatial evolution process. The fourth part concludes and proposes the corresponding policy suggestions. The final section discusses the limitations of the paper and future research prospects.

The research framework of this paper is as following Figure 1.

## 2. Research Design

### 2.1. Research Method

#### 2.1.1. The Super-SBM Model Using Undesirable Output

The efficiency of green technology innovation represents the innovation ability of clean production technology. A less unexpected output of the regional environment can decrease the environmental capacity consumption of technological innovation, and can strengthen the innovation ability of green technology. The regional innovation system is highly complex, involving a variety of inputs and outputs. Therefore, this paper uses DEA as an evaluation tool for the efficiency of green technological innovation. A commonly used tool for measuring the efficiency of green technology innovation is known as data envelopment analysis (DEA). However, most traditional DEA models are radial, and angle measurements and the attributes that can be processed for undesirable outputs are insufficiently considered. To overcome this defect, this paper uses the non-radial and non-angle super-SBM model proposed by Tone [27] to measure the industrial green technology innovation efficiency of the Yangtze River Economic Belt. The super-SBM model can calculate the decision-making unit with an efficiency value greater than 1, effectively reflecting industrial green technology innovation efficiency in different regions of the Yangtze River Economic Belt. 

Each decision-making unit contains three vectors: input X, expected output $Y^{g}$, and unexpected output $Y^{b}$. If m unit of input is consumed, *s*_1_ and *s*_2_ units of expected production and unexpected output will be generated. Therefore, these three vectors are respectively expressed as $x\in R^{m}$, $y^{g}\in R^{s_{1}}$, and $y^{b}\in R^{s_{2}}$. The matrix is defined as follows:(1)$$\begin{array}{l} X=\left[x_{1},x_{2}\cdots x_{n}\right]\in R^{m\times n}\\ Y^{g}=\left[y_{1}^{g},y_{2}^{g}\cdots y_{n}^{g}\right]\in R^{s_{1}\times n}\\ Y^{b}=\left[y_{1}^{b},y_{2}^{b}\cdots y_{n}^{b}\right]\in R^{s_{2}\times n} \end{array}$$

$x_{i}>0,y_{i}^{g}>0,y_{i}^{b}>0\left(i=1,2\cdots n\right)$. The super-SBM model based on unexpected output can be expressed as:(2)$$\begin{array}{l} \rho^{*}=\min\frac{\frac{1}{m}\sum_{i=1}^{m}\frac{\bar{x_{i}}}{x_{i0}}}{\frac{1}{s_{1}+s_{2}}\left(\sum_{r=1}^{s_{1}}\frac{{\bar{y}}_{r}^{g}}{y_{r0}^{g}}+\sum_{u=1}^{s_{2}}\frac{{\bar{y}}_{r}^{g}}{y_{l0}^{b}}\right)}\\ s.t.\left\{ \begin{array}{l} \bar{x}\geq \sum_{j=1,\neq 0}^{n}\lambda_{j}x_{j},j=1,\cdots ,m\\ {\bar{y}}^{g}\leq \sum_{j=1,\neq 0}^{n}\lambda_{j}y_{j}^{g},r=1,\cdots ,s_{1}\\ {\bar{y}}^{b}\geq \sum_{j=1,\neq 0}^{n}\lambda_{j}y_{j}^{b},l=1,\cdots ,s_{2}\\ \bar{x}\geq x_{0},{\bar{y}}^{g}\leq y_{0}^{g},{\bar{y}}^{b}\geq y_{0}^{b},\lambda \geq 0,\sum_{j=1,\neq 0}^{n}\lambda_{j}=1 \end{array}\right. \end{array}$$

$\rho^{*}$ is the green technology innovation efficiency value (the larger the value, the higher the efficiency), $y^{g}$ is the expected output, $y^{b}$ is the undesired output, and λ is the weight vector. When the efficiency value $\rho^{*}$ is greater than or equal to 1, it means that the decision-making unit is in a valid state, and when $\rho^{*}$ is between 0 and 1, it means that the decision-making unit is in an invalid state.

#### 2.1.2. The Global Malmquist–Luenberger Index

The efficiency value of industrial green technology innovation can be obtained using the above super-SBM model, which can compare the efficiency of different regions in a certain period. It is a static analysis and cannot effectively reflect the dynamic change characteristics of industrial green technology innovation in sequential evolution. Therefore, this paper draws on Oh’s [28] research and introduces the GML index decomposition model to dynamically analyze the efficiency value of industrial green technology innovation in the Yangtze River Economic Belt. Based on the super-SBM distance function, and according to the GML index method constructed by Pastor [29], the GML index from t period to *t* + 1 period is:(3)$$\begin{array}{l} GML_{t}^{t+1}\left(x^{t},y^{t},b^{t},x^{t+1},y^{t+1},b^{t+1}\right)=\frac{E^{g}\left(x^{t+1},y^{t+1},b^{t+1}\right)}{E^{g}\left(x^{t},y^{t},b^{t}\right)}\\ =\frac{E^{t}\left(x^{t+1},y^{t+1},b^{t+1}\right)}{E^{t}\left(x^{t},y^{t},b^{t}\right)}\times \left[\frac{E^{g}\left(x^{t+1},y^{t+1},b^{t+1}\right)}{E^{t+1}\left(x^{t+1},y^{t+1},b^{t+1}\right)}\times \frac{E^{t}\left(x^{t},y^{t},b^{t}\right)}{E^{g}\left(x^{t},y^{t},b^{t}\right)}\right]\\ =EC_{t}^{t+1}\times TC_{t}^{t+1} \end{array}$$

When the $GML_{t}^{t+1}$ index is less than 1, the efficiency of industrial green innovation will decline from period *t* to period *t* + 1. When the $GML_{t}^{t+1}$ index equal to 1, there will be no change. When the $GML_{t}^{t+1}$ index is than 1, there will be growth. $EC_{t}^{t+1}$ represents the efficiency from period t to period *t* + 1. The change index reflects the “catch-up effect” of the decision-making unit on the production frontier. $EC_{t}^{t+1}$ greater than (less than) 1 indicates that the efficiency will improve (decreased) in two periods. $TC_{t}^{t+1}$ indicates the technological change index from period t to period *t* + 1, reflecting the shift effect of the internal production frontier. $TC_{t}^{t+1}$ greater (less than) 1 indicates technological progress (regression) over two periods.

#### 2.1.3. Dagum Gini Coefficient

The coefficient of variation, the Theil index, and the Gini coefficient are often used to measure the regional differences, and the Theil index is often used to decompose the regional differences. However, the traditional Gini coefficient and Theil index struggle to solve the cross overlap of sample data and the source of regional differences. The Gini coefficient decomposition method considers the distribution of subgroup samples, solves the problem of cross overlap between sample data and the source of overall regional differences, and overcomes the shortcomings of the traditional Gini coefficient and Theil coefficient. It has been widely used in various fields of the economy [30]. This paper uses the Dagum Gini coefficient and its subgroup decomposition method to study the regional differences and sources of regional differences in industrial green technology efficiency in the Yangtze River Economic Belt. The formula is as follows: (4)$$G=\frac{\sum_{j=1}^{k}\sum_{h=1}^{k}\sum_{i=1}^{n_{j}}\sum_{r=1}^{n_{k}}\left|y_{ji}-y_{hr}\right|}{2n^{2}\bar{y}}$$

k=3 represents the upper, middle, and lower reaches of the Yangtze River Economic Belt, while j,h represents the three regions’ different regions. n represents the total number of cities in the Yangtze River Economic Belt. In this paper, the value is 110. $n_{j},n_{h}$ is the number of towns in the j,h region, $y_{ji},y_{hr}$ is the city i,r efficiency of industrial green technology innovation in the j,h region, and $\bar{y}$ is the average efficiency of industrial green technology innovation in the Yangtze River Economic Belt. The Gini coefficient G can be decomposed into intra-regional gap contribution $G_{w}$, inter-regional net value gap contribution $G_{nb}$, and hypervariable density $G_{t}$, satisfying $G=G_{w}+G_{nb}+G_{t}$. 

#### 2.1.4. Kernel Density Estimation

To intuitively and vividly reveal the distribution pattern, change trend, and elasticity of the spatial–temporal evolution trend of industrial green innovation efficiency in the Yangtze River Economic Belt, this paper introduces the kernel density function to describe the overall shape of the innovation efficiency part. It compares the dynamic changes in the efficiency distribution in different periods [31]. Kernel density estimation is one of the non-parametric methods that can effectively investigate the distribution form of random variables through the continuous density function and then complete the kernel density estimation of samples [32]. Because of its strong robustness and weak model dependence, it has been widely used to analyze the dynamic evolution of sample differences and distribution between regions. Assuming that the density function of random variable X is f(x), the probability density expression at point *x* is:(5)$$f\left(x\right)=\frac{1}{nh}\sum_{t=1}^{n}K\left(\frac{x-x_{i}}{h}\right)$$

Among them, n is the number of observations, K(⋅) is the kernel density function; h is the bandwidth, the size of which determines the accuracy of the kernel density estimation and the smoothness of the density map; x is the independently distributed sample observations; and $x_{i}$ is the mean. The kernel density function has a Gaussian kernel, an Epanechnikov kernel, a triangular kernel, a quartic kernel, and other forms [33]. Drawing on the existing research [34], this paper uses the Gaussian kernel function to make estimations. The formula is as follows:(6)$$K\left(x\right)=\frac{1}{\sqrt{2\pi }}\exp\left(-\frac{x^{2}}{2}\right)$$

#### 2.1.5. Spatial Markov Chain

The nuclear density estimation cannot reflect the change in the relative position of industrial green innovation efficiency and the possibility of change. Therefore, to analyze the spatial evolution process of green industrial innovation in the Yangtze River Economic Belt, this paper first uses the Markov chain to construct a transfer matrix to explore innovation efficiency’s spatial and temporal evolution between different regions. The industrial green innovation efficiency is divided into other states, and a Markov transfer probability matrix is obtained [35]. Then, according to the level of innovation efficiency, according to Tang’s [36] research, the transfer direction is divided into four types: low (I), medium–low (II), medium–high (III), and high (IV). However, the traditional Markov chain needs to set the study area as independent individuals, and there is no spatial connection between them. It is a random sequence method with discrete time and state [37]. Then, the spatial lag factor ($lag=\sum_{i=1}^{n}\omega_{ij}x_{i}$, $\omega_{ij}$ is the adjacency matrix of two regions, and $x_{i}$ is the value of regional innovation efficiency) is introduced based on the traditional Markov chain. Considering the adjacent spatial lag in the region i in the initial year as the condition, the conventional k×k order Markov transition probability matrix is transformed into k conditional transition probability matrices k×k. The probability of industrial green technological innovation efficiency transfer from type i to type j under the adjacent spatial lag type k in period t is investigated [38]. The state of inter-regional independence is overcome to reveal the influence of spatial correlation on the dynamic evolution of industrial green innovation efficiency. 

### 2.2. Construction of the Evaluation Index System

The calculation of green technology innovation efficiency includes the rational analysis of “innovation efficiency” and considers the “green” factor. Therefore, this paper draws on the existing research results [39,40]; adds the concept of green to the original input–output index system; takes into account the rationality and scientific principle of index selection; and constructs the index system, including the input, the expected output, and the undesirable output (as shown in Table 1) to systematically quantify the efficiency of industrial green technology innovation in the Yangtze River Economic Belt, making the calculation results more accurate. Indicators are described below.

Input variables: Considering that the entire process of industrial green technology innovation has higher requirements for general human and capital investment and requires a large number of innovative elements of investment, select R&D personnel investment, R&D funding, and industrial energy investment act as three secondary indicators. The total amount of R&D full-time personnel workload and non-full-time personnel conversion workload in the industry is reflected by the full-time equivalent of R&D personnel. R&D expenditure mainly refers to new product development and technological transformation funds. Industrial energy input is represented by the total energy consumption, which is the consumption of various types of energy in the industry.

Expected output: The expected output indicators are mainly divided into new product sales revenue, the number of new product development projects, effective invention patents, and total sale values. The sales revenue of new products reflects the transformation ability of green technological innovation achievements, which can bring certain economic benefits to the whole industry and represent the marketization level of innovation achievements. The number of new product development projects and the number of effective invention patents embody industrial R&D innovation capability. Generally speaking, more unique product development projects are will lead to higher enterprise investment, more resource factors, and more innovative achievements.

Unexpected output: The key to the efficiency evaluation of industrial green technology innovation is to consider the reduction in energy consumption and pollution. Pollutants are undesirable outputs, mainly divided into industrial wastewater emissions, industrial sulfur dioxide emissions, and solid waste production. Among them, industrial wastewater emission is primarily represented by chemical oxygen demand (COD) emission and ammonia nitrogen emission.

### 2.3. Data Sources

The Yangtze River Economic Belt, which spans three major regions in China, is one of the “three effective strategies” implemented by the central government. It is an inland river economic belt with global influence, a coordinated development zone of east–west interaction and cooperation, and an opening-up zone to the outside world comprehensively promoted along the coastal river. It is also an advanced demonstration zone of ecological civilization construction. The Yangtze River Economic Belt covers Shanghai, Jiangsu, Zhejiang, Anhui, Jiangxi, Hubei, Hunan, Chongqing, Sichuan, Guizhou, Yunnan, and other 11 provinces and cities, covering an area of about 20.523 million km^2^, accounting for 21.4% of the country. According to the upper, middle, and lower reaches, the lower reaches include Shanghai, Jiangsu, Zhejiang, and Anhui provinces, with an area of about 350.3 million km^2^, accounting for 17.1% of the Yangtze River Economic Belt; the middle reaches include Jiangxi, Hubei, and Hunan provinces, with an area of 5,646,600 km^2^, accounting for 27.5% of the Yangtze River Economic Belt; the upper reaches include Chongqing, Sichuan, Guizhou, and Yunnan provinces and cities, with an area of about 11.374 million km^2^, accounting for 55.4% of the Yangtze River Economic Belt. While implementing the spatial layout as the carrier of the practical orientation and tasks of the Yangtze River Economic Belt, the pattern of “one axis, two wings, three poles, and multiple points” is proposed.

The research sample selected in this paper is the panel data of 110 prefecture-level cities in the Yangtze River Economic Belt (including 33 cities in the upper reaches, 36 cities in the middle reaches, and 41 cities in the lower reaches), and the period is 2006–2020. The data needed for the above index system mainly come from the “China Industrial Statistics Yearbook”, “China Energy Statistics Yearbook”, “China Science and Technology Statistics Yearbook”, and the statistical yearbooks and statistical bulletins of the corresponding years of each province and city. Some missing values are filled using the interpolation method. The patent data of invention comes from the State Patent Office, and the “three wastes” in the unexpected output mainly come from the Guotaian database.

## 3. Analysis of Empirical Results

### 3.1. Static Analysis of Industrial Green Technology Innovation Efficiency

With the help of MaxDEA Ultra8.0 software, this paper uses the super-SBM model based on unexpected output to measure the industrial green technological innovation efficiency of 110 cities in the Yangtze River Economic Belt. To facilitate the analysis and comparison of the temporal variation characteristics and interregional differences in innovation efficiency in different regions, the Yangtze River Economic Belt is divided into the upper, middle, and lower reach as Figure 2.

Overall, during the study period, the industrial green innovation efficiency of the Yangtze River Economic Belt showed a trend of “N” type, which rose slowly first and then decreased and then increased, reaching the lowest point in 2009. The possible reason is that the outbreak of the financial crisis in 2008 had a significant impact on China’s import and export trade, and the turbulence of the financial market has seriously affected the marketization of China’s green technology innovation achievements, which led to a decline in industrial green technology innovation efficiency. Subsequently, the Yangtze River Economic Belt has increased the construction of innovative infrastructure, especially the “twelfth five-year” plan, which puts forward the concept of green development and actively introduces innovative talents, and the government gives preferential policies in green innovation, which steadily increases the efficiency of green technology innovation. However, from the average point of view, the efficiency of green industrial innovation in the Yangtze River Economic Belt is below 1, which does not reach an effective state, indicating that the efficiency of industrial green technology innovation in the Yangtze River Economic Belt still has room for considerable improvement.

In terms of sub-regions, the green technological innovation efficiency of the lower region is the strongest, followed by the middle region, and the upper region is the weakest. The efficiency level of green technology innovation in the three areas has evolved from a “low-efficiency small gap” to a “high-efficiency large gap”. The gap has gradually narrowed since 2016. The main reason is that the downstream region, due to its unique geographical location and resource endowment advantages, gathers a large number of scientific and technological innovation talents, has muscular economic strength, and has a relatively perfect industrial system of scientific and technological innovation, which is conducive to the transformation of technological innovation achievements. As a traditional manufacturing base in China, the middle reaches gather many high-energy-consuming and high-pollution industries, such as iron, steel, and chemical industries, with low technological content. The resource allocation rate is relatively low, and the innovation ability of green technology is weak. The upstream region lacks resources, and the overall economic development level is low. It tends to pursue rapid economic development at the expense of the environment, and the environmental regulation is weak, possibly leading to low efficiency of green technology innovation. However, with the “13th Five-Year Plan” in 2015, and with green development as the first of five development concepts, green growth has brought new opportunities to the upstream and middle reaches of the region, promoting scientific and technological innovation and technological change, forcing industrial enterprises to achieve structural transformation and improve the efficiency of green innovation, and thus rapidly increasing the efficiency of green technology innovation. The gap in the downstream region has gradually narrowed. The upstream region has excellent potential for development and presents an indigenous green catch-up effect, and its green innovation efficiency has gradually exceeded the middle reaches.

To reveal the spatial and temporal evolution trend of industrial green innovation efficiency of 110 cities in the Yangtze River Economic Belt, this paper selects four-time nodes in 2006, 2010, 2015, and 2020. It uses Arcgis10.3 software to classify the industrial green technology innovation efficiency of the Yangtze River Economic Belt into high efficiency (≧0.8), high efficiency (0.6, 0.8), low efficiency (0.4, 0.6), and low efficiency (≦0.4).

It can be seen from Figure 3 that the industrial green technological innovation efficiency of 110 cities in the Yangtze River Economic Belt shows a trend that the cities at a low level gradually move towards a low level or a high level. In 2006, the towns were mainly low-level. With time, the cities at a low level gradually decrease. In 2020, the industrial green technological innovation efficiency of Baoshan City, Lincang City, and Simao City will be below 0.4, while the cities at a high level will gradually increase. By 2020, the towns with efficiency greater than 0.8 will rise to 26, accounting for 23.6% of the total. This shows that these cities have increased investment in green innovation in economic development, significantly improving efficiency. On the whole, the industrial green technological innovation efficiency of the Yangtze River Economic Belt presents a typical “center-periphery” dual spatial structure, indicating that the green technology innovation ability of coastal and central cities along the river is significantly higher than that of other cities. This means that cities with high innovation efficiency are mainly concentrated in the Yangtze River Delta region and some provincial capital cities, such as Wuhan, Changsha, Kunming, and Chengdu, as well as their neighborhoods. The agglomeration effect is pronounced, which plays a leading role in improving the innovation efficiency of local and adjacent regions. It is an essential part of promoting the green development of the Yangtze River Economic Belt.

### 3.2. Dynamic Analysis of Industrial Green Technology Innovation Efficiency

The change in efficiency is a dynamic and continuous evolution process. The above super-SBM model is based on non-radial and non-angle analyses of the change and spatial differentiation characteristics of industrial green technology innovation efficiency in the Yangtze River Economic Belt from a static perspective. To further compare the change in the dynamic efficiency of industrial green technology innovation from a vertical standpoint in the period, this paper selects the GML index (see Table 2) to measure and decompose the industrial green technology innovation efficiency in the Yangtze River Economic Belt from 2006 to 2020 and analyzes its primary power sources.

It can be seen from the above table that the average GML index of industrial green technological innovation efficiency in the Yangtze River Economic Belt during the study period is greater than 1, and that technological progress increases by 7.1% on average. Technical efficiency decreases by 1.5%, indicating that the overall efficiency of industrial green technological innovation shows an upward trend, and technological progress is the primary power source to promote its growth. The main reason is that technological progress refers to the continuous improvement, development, and innovation of science and technology, a research and development activity aimed at bringing a new face to production and life. Technological progress helps to promote industrial transformation, as well as upgrading and adjusting energy structure, thereby improving resource utilization and energy consumption and promoting innovation efficiency. However, there are fluctuations in individual years. From the changes in each year, the GML index of industrial green technological innovation efficiency in 2007–2008, 2008–2009, and 2015–2016 is less than 1, which may be due to the impact of the global financial crisis. The high-leverage and high-risk development model intensifies the difficulty of transforming innovation achievements in the Yangtze River Economic Belt. Through the observation of TC and EC in 2007–2008 and 2008–2009, it can be seen that technical efficiency is the main reason for a decline in the GML index. Subsequently, China’s industrial economy recovered rapidly, and the research and development of technological innovation increased. By actively carrying out industrial transformation and upgrading and structural adjustment, the efficiency of industrial green technology innovation has increased year by year. However, by 2015, the “green” and “ecological” process and results of economic activities are the main contents and ways of green development for the first time in the Fifth Plenary Session of the 18th Central Committee of the Communist Party of China. Local governments have increased investment in innovation. However, due to a lack of new technology development and R&D management experience, it is not easy to effectively improve technology quickly; therefore, the efficiency of industrial green technology innovation in the Yangtze River Economic Belt has declined briefly. From a regional perspective, because of China’s “East before West” development strategy, the average value of technological progress shows the spatial distribution characteristic of a gradual decrease from the downstream region to the upstream region. The average weight of technical efficiency in the upstream region is the largest, followed by the downstream and middle reaches, indicating that although the economic development of the upstream region lags behind that of the downstream area, the rationalization of industrial structure can realize the effective utilization of resources and increase the transformation efficiency of innovation achievements.

### 3.3. Regional Difference and Decomposition of Industrial Green Technological Innovation Efficiency in the Yangtze River Economic Belt

To describe the regional differences and sources of industrial green technical efficiency in the Yangtze River Economic Belt, this paper uses the Gini coefficient and its subgroup decomposition method proposed by Dagum to decompose China’s industrial environmental technical efficiency under the constraints of resources and environment from 2006 to 2020. The calculation results are shown in Table 3.

The results show that during the sample period, the overall Gini coefficient decreased from 0.484 in 2006 to 0.353 in 2020, showing a down–up–down W-shaped fluctuation. The level of high-quality economic development in the upstream, midstream, and downstream regions shows similar changes, and the Gini coefficient in the downstream region is always higher than that in the central and upstream regions. The downstream regions mainly include Jiangsu, Zhejiang, Anhui, and Shanghai. These regions have good overall natural conditions and convenient transportation, conducive to population settlement and economic development. However, the region includes developed cities (such as Shanghai, Nanjing, Suzhou, and Hangzhou) and underdeveloped cities (such as Fuyang, Suzhou, and Yancheng). There are significant differences in the intensity of economic development, human capital, and R&D investment; thus, the efficiency of industrial green technology innovation in the downstream areas is quite different. The Gini coefficient between the middle and upper reaches had little difference before 2014; however, after 2014, the Gini coefficient in the intermediate spaces showed a rapid downward trend. The reason is that the implementation of the Central Rise Plan has made remarkable achievements in the economic and social development of the central region, and has greatly improved the level of economic development. Moreover, the government has increased its policy support for the main area, showing an inevitable decline in the Gini coefficient within the region.

The regional differences in industrial green technology innovation levels in the Yangtze River Economic Belt show a cyclic trend of “decline-rise-fall”, but the fluctuation is insignificant. The difference in high-quality economic development levels between the upstream and downstream regions is the largest, and the Gini coefficient varies between 0.186 and 0.679. The regional difference between the midstream and downstream regions is moderate, and the Gini coefficient fluctuates between 0.122 and 0.610. The difference between the upstream and midstream is minimal, and the Gini coefficient fluctuates between 238 and 0.508. The contribution rate of regional difference is basically in a stable state, and the regional difference shows a downward trend. From 41.35% in 2006 to 31.36% in 2020, the contribution rate of excess density increased from 27.65 % in 2006 to 38.26 in 2020. Regional differences and excess density are the primary sources of industrial green technology innovation efficiency differences in the Yangtze River Economic Belt. The main reason is that the upstream, midstream, and downstream areas have differences in regional position, resource endowment degree, science and technology education culture, and many other aspects; thus, the downstream continues to optimize the innovation cluster. However, the upstream area has insufficient innovation due to the weak economy and lack of talent. The economic development of the middle reaches has long been mainly driven by domestic demand. The lack of innovative resources has been insignificant in connecting the east and the west.

### 3.4. Kernel Density Estimation of Industrial Green Technology Innovation Efficiency

To ensure the accuracy of the non-parametric estimation results of nuclear density, this paper selects 2006, 2010, 2015, and 2020 as observation samples. The Gaussian kernel density function (see Figure 4) is used to make the two-dimensional convergence map of industrial green technology innovation efficiency in the Yangtze River Economic Belt. 

From the peak position, the kernel density function center of industrial green technological innovation efficiency in the Yangtze River Economic Belt shows a right-shift trend during the study period, indicating that its innovation efficiency level improved to a certain extent, which is consistent with the above conclusions. In particular, the lower reaches of the Yangtze River actively promote green development policies, and the green innovation efficiency is better than that of other regions. From the distribution pattern, the peak height of nuclear density estimation shows a trend of “down-up-down”, indicating that the absolute difference between industrial green technology innovation efficiency regions in the Yangtze River Economic Belt first expands and then decreases and then expands. From the perspective of distribution elasticity, there is a right tail phenomenon in the nuclear density estimation curve, which indicates that the gap between industrial green technology innovation efficiency in the Yangtze River Economic Belt gradually increases in high-level and low-level areas. From the number of peaks, there was a “double peak” phenomenon of a central peak and a side peak in 2006, but the characteristics of the double peak gradually disappeared in the later period. The nuclear density estimation curve was mainly a “single peak”, indicating that the efficiency of industrial green technology innovation in the Yangtze River Economic Belt during the study period was initially polarized. After 2010, this polarization disappeared and gradually converged to a single equilibrium point.

### 3.5. Spatial Markov Chain Analysis of Industrial Green Technology Innovation Efficiency

The evolution trend and non-equilibrium characteristics of industrial green technology innovation efficiency in the Yangtze River Economic Belt can be seen through the nuclear density estimation. Still, it cannot describe the dynamic change in each region’s relative position within the form distribution of industrial green technology innovation efficiency and its transfer probability. Therefore, this paper uses the Markov chain to explore the dynamic evolution of industrial green technology innovation efficiency distribution. Referring to the relevant literature research [41], the green technology innovation efficiency is divided into four grades: a low level (I) below 25%, a medium and low level (II) between 26 and 50%, a medium and high level (III) between 51 and 75%, and a high level (IV) above 75%. The probability transfer matrix of the traditional Markov chain and the spatial Markov chain in the dynamic evolution process of industrial green innovation efficiency distribution in the Yangtze River Economic Belt is obtained, as shown in the following Table 4.

From the traditional Markov chain probability transfer matrix, it can be seen that the efficiency level of industrial green technology innovation in the Yangtze River Economic Belt has the phenomenon of “club convergence”, and the transfer probability value on the diagonal is relatively high, rather than the transfer probability value on the diagonal is low. This shows that the possibility of maintaining the original level of innovation efficiency from the *t* period to the *t* + 1 period is significant and has specific stability. During the study period, the probability of innovation efficiency maintaining at a high level and a low level is the highest, 97.1% and 80.6%, respectively, indicating that the industrial green technology innovation efficiency of the Yangtze River Economic Belt presents an evident convergence trend. The “Matthew effect” is noticeable, which means that regions at a high level and a low level in period *t* maintain the original level at a high probability in period *t* + 1. The probability value on the right side of the non-diagonal is greater than the probability value on the left side, indicating that the probability of upward transfer of innovation efficiency is greater than that of downward transfer, which is consistent with the above research on the significant improvement of industrial green innovation efficiency in the Yangtze River Economic Belt. By analyzing the probability value on the non-diagonal, there is a possibility of leapfrog transfer in the innovation efficiency. For example, the probability of low-level transfer to a medium–high level is 1.3%, and the likelihood of a medium–low-level transfer to a high level is 1.8%. Still, its value is far less than the probability of transfer to adjacent levels. For example, the probability of a medium–low-level transfer to a high level is 30.6%. This shows that the efficiency of industrial green technology innovation in the Yangtze River Economic Belt has a strong path dependence. It is challenging to achieve leapfrog development in a short period. 

Compared with the traditional Markov chain probability transfer matrix and spatial Markov chain probability transfer matrix, it can be seen that the transfer of industrial green technology innovation efficiency in the Yangtze River Economic Belt shows a specific spatial dependence under the action of geographical spatial effect. Specifically, (1) the probability transfer matrix of innovation efficiency is different under the influence of different types of neighborhood space lag. Without considering the neighborhood space lag, the probabilities of four innovation efficiency levels to maintain stability are 80.6%, 66%, 74.2%, and 97.1%, respectively. Under the condition of considering different neighborhood space lags, the probabilities of low-level innovation efficiency to maintain stability are 84.7%, 74.9%, 70.2%, and 63.7%, respectively. The medium–low-level innovation efficiency probabilities of maintaining stability are 72%, 77.3%, 79.3%, and 64.8%, respectively. The medium–high-level innovation efficiency probabilities of maintaining stability are 70.5%, 69%, 76.3%, and 69.4 respectively. The high-level innovation efficiency probabilities of maintaining stability are 91.6%, 93.5%, 96.9%, and 100%, respectively. Compared with the traditional Markov chain probability transfer matrix, the low-level innovation efficiency greatly influences the neighborhood space lag type. The probability of maintaining stability decreases from 80.6% to 63.7%. This shows that the industrial green technology innovation efficiency of the Yangtze River Economic Belt has the “club convergence” effect, which is affected by the positive “spillover effect” of the high-level region. Low-level areas have the potential to break through low-level locking. (2) The higher level of neighborhood space lag type has a particularly positive effect on the innovation efficiency of surrounding areas, which makes it easier for low-level cities to transfer to higher-level cities. For example, low-level city type I moving to medium–low-level city type II is 18.1% without considering the neighborhood space lag type. In comparison, the likelihood of low-level city type I transferring to medium–low-level city type II is 12.7%, 25.1%, 28.5%, and 31.8%, respectively. This indicates that high-level cities have a radiation-driven effect on the innovation efficiency of neighboring regions. Thus, there is a positive “spillover effect”. The low level of neighborhood space lag type has a particularly inhibitory impact on the innovation efficiency of the surrounding areas, which makes it easier for high-level cities to transfer to lower-level cities. For example, in the case of neighborhood space lag type II, the probability of transfer from high-level city type III to low-level city type II is 3.5%. In contrast, in the case of neighborhood space lag type I, the probability value rises to 16.4%, indicating that cities with low innovation efficiency have a particularly negative effect on the development of surrounding municipalities. Therefore, there is a negative “spatial spillover” effect. (3) Under the influence of a high level of neighborhood spatial lag, the possibility of the region’s transfer to lower-level cities will be reduced. For example, when the type of neighborhood spatial lag is I, the probability of transferring medium–high-level city type III to lower-level city type II is 16.4%. When the neighborhood spatial lag is II, its value is reduced to 3.5%. When the type of neighborhood spatial lag is III and IV, its value is 0, indicating that with an increase in the type of neighborhood spatial lag can increase the difficulty of cities with high innovation efficiency transferring to lower-level cities. (4) In the above table showing probability transfer matrix, the values on both sides of the adjacent diagonal are more significant, and the values far from the diagonal are smaller. For example, in the case of neighborhood space lag type I, the probability of transfer from low-level city type I to medium–low-level type II is 12.7%, which is much larger than the probability of transfer from low-level city type I to medium–high-level type III by 2.6%, similar to the other types. This shows that the transfer between different innovation efficiency levels has strong path dependence. It is easier to transfer between adjacent level levels. The probability of leapfrog transfer is small, resulting from the traditional Markov chain probability transfer matrix.

## 4. Conclusions and Suggestion

Based on the panel data of 110 cities in the Yangtze River Economic Belt from 2006 to 2020, this paper first uses the non-radial and non-angle super-SBM model and the GML index decomposition model to calculate the efficiency of industrial green technology innovation in the Yangtze River Economic Belt. It compares and analyzes the efficiency of industrial green technology innovation in the upper, middle, and lower reaches of the Yangtze River Economic Belt from static and dynamic aspects. At the same time, the exploratory spatial data analysis (ESDA) is used to explore the spatial differentiation characteristics and the evolution law of innovation efficiency at the urban level. Then, by using the Dagum Gini coefficient and its subgroup decomposition, the kernel density estimation method, and spatial Markov chain, the regional differences and convergence characteristics of industrial green technology innovation efficiency in the Yangtze River Economic Belt and the dynamic evolution law of its distribution are analyzed. The following conclusions are drawn. (1) Through the calculation of the super-SBM model considering the unexpected output, it can be seen that the efficiency of industrial green technological innovation in the Yangtze River Economic Belt presents an “N” trend of a slow increase first and then a decrease and then an increase on the whole, and presents an unbalanced feature of ‘high in the east and low in the west’ in space, similar to the study of Xu et al. [42]. However, based on the urban level, this paper concludes that the efficiency space of green innovation presents a typical “center-periphery” dual structure. The high-level cities with high innovation efficiency are mainly concentrated in the Yangtze River Delta region, Wuhan, Changsha, Kunming, Chengdu, and some other provincial capital cities nearby, and the agglomeration effect is remarkable. (2) Through the dynamic analysis of industrial green technology innovation efficiency in the Yangtze River Economic Belt, it is known that the average GML index of innovation efficiency is greater than 1, the average technical progress is increased by 7.1%, and the technological efficiency is decreased by 1.5%, indicating that technical progress is the main driving force for efficiency growth, similar to the study of Tian et al. [43]. However, this paper compares the decomposition results of TC and EC in different regions. Technological progress presents the spatial distribution characteristics of a gradual decline from downstream to upstream. Technical efficiency is the largest upstream region, followed by the downstream and midstream areas. (3) The results of the Dagum Gini coefficient calculation and decomposition show spatial and temporal differences in industrial green technology innovation in the upper, middle, and lower reaches of the Yangtze River Economic Belt. The overall Gini coefficient and the Gini coefficient of inter-regional differences in the region show a fluctuating state of decline–rise–decline. Among them, the innovation level between the upstream and the downstream is quite different, and the regional difference between the upstream and the middle reaches is minimal. Inter-regional differences and excess density are the primary sources of the overall difference in the level of industrial green technology innovation in the Yangtze River Economic Belt. (4) During the study period, the kernel density curve center of industrial green technology innovation efficiency in the Yangtze River Economic Belt shows a right-shift trend. The peak height shows the change characteristics of “down-up-down”. This shows that the absolute difference in industrial green technology innovation efficiency in the Yangtze River Economic Belt continues to improve. The absolute difference between regions shows the “expansion-reduction-expansion” trend. With the evolution of time, the number of peaks gradually changes from the initial “double peaks” to “single peaks”, indicating that the polarization of industrial green technology innovation efficiency in the Yangtze River Economic Belt gradually disappears and converges to a single equilibrium point. (5) According to the Markov chain probability transfer matrix, industrial green technology innovation efficiency transfer in the Yangtze River Economic Belt shows a specific spatial dependence. The transfer of innovation efficiency at different levels mainly occurs between adjacent clubs, and it is challenging to achieve leapfrog development in a short time, similar to the study of Wang et al. [44]. However, this article also suggests that different types of neighborhood space lag have different effects on innovation efficiency. The higher level of neighborhood space lag environment has a radiation-driven effect on the surrounding cities. The lower level of neighborhood space lag environment negatively affects the surrounding cities. To further improve the level of industrial green technology innovation in the Yangtze River Economic Belt, the following policy suggestions are put forward.

(1) Overall, it is important to understand the spatial transmission mechanism of industrial green technology innovation efficiency in the Yangtze River Economic Belt. First of all, due to the differences in resource endowments and economic development in the upper, middle, and lower areas, it is necessary to actively encourage the transfer of green industries in the lower areas to the middle and upper areas. At the same time, it is essential to strengthen infrastructure construction in the middle and lower areas, create a good business environment, strengthen technical exchanges and talent flows in different regions while undertaking Lower industries, break technical barriers, and realize cross-regional coordinated development. Secondly, the advantages of core cities, as well as cities with high innovation efficiency in the “siphon” adjacent areas of the capital with talent and technology, should be addressed to increase the “nurturing” effect on adjacent areas, enhance the spatial radiation of green innovation, and ensure collaborative development of innovation efficiency.

(2) From the research results, technological innovation plays a crucial role in improving efficiency. Therefore, it is necessary to strengthen the core development strategy driven by innovation, implement differentiated technological innovation policies according to local conditions, encourage enterprises to increase capital and human input in technological innovation and development, improve resource allocation efficiency, and accelerate the market conversion rate of green innovation achievements. Secondly, the adjustment of the industrial structure should be optimized. The lower areas should vigorously develop technology-intensive industries, optimize the energy structure, and improve resource utilization efficiency. In undertaking lower industrial transfer, the middle and upper reaches should eliminate backward production capacity, cultivate new growth poles of economic development, and improve green innovation.

(3) In order to give liberty to the government’s policy guidance and support their role in industrial green technology innovation, we should first create an excellent green innovation environment and a fair competition system, adhere to the leading role of the market, weaken the excessive government intervention and monopoly, and protect the green innovation of enterprises. Secondly, while strengthening environmental regulation, it is also necessary to give policy optimization to green technology innovation enterprises. At the same time, it is essential to avoid adverse and vicious competition in the market, strictly abide by the red line of environmental protection, and take green technology innovation as the long-term development strategy of the Yangtze River Economic Belt and even the whole country.

(4) In addition, promoting the reform of decentralization, management, and service of environmental governance; flexibly promoting ecological protection policies; improving the effectiveness of environmental protection policies; and building a unified environmental early warning and monitoring system can be carried out to encourage urban environmental information sharing in the upper, middle, and lower reaches. Furthermore, it is important to promote the green transformation and upgrading of industrial structure, cultivate and expand green emerging capacity, and enhance the core technical reserves of green emerging power. Finally, constructing regional green innovation network, implementing green innovation pilot demonstration project, encouraging central cities and surrounding cities to strengthen green technology cooperation, striving to break regional barriers, building regional green innovation network platforms, and promoting the coordinated improvement in the green technology innovation ability of cities in the Yangtze River Economic Belt are also useful and important strategies.

### Limitations and Future Research

The shortcomings of this paper are as follows. In the selection of research samples, the data of individual cities are accurate in some years, such as the data of Bijie City in 2008, 2014, and 2015, which are calculated using interpolation method. With regards to choosing indicators, mainly in reference to China’s choice of this research index, there is a lack of comparison with other countries. In some analyses, given the length of the article, there is a lack of detailed analysis of different cities in the upper, middle, and lower reaches of the Yangtze River. Therefore, it is hoped that more comprehensive data can be collected in future research. In terms of selecting indicators, relevant literature in China and abroad can be combined to make a scientific choice. To analyze the spatial–temporal evolution mechanism in more detail, the upstream, midstream, and downstream of the Yangtze River Economic Belt can be taken as the research samples separately.

## Figures and Tables

**Figure 1 ijerph-19-06361-f001:**
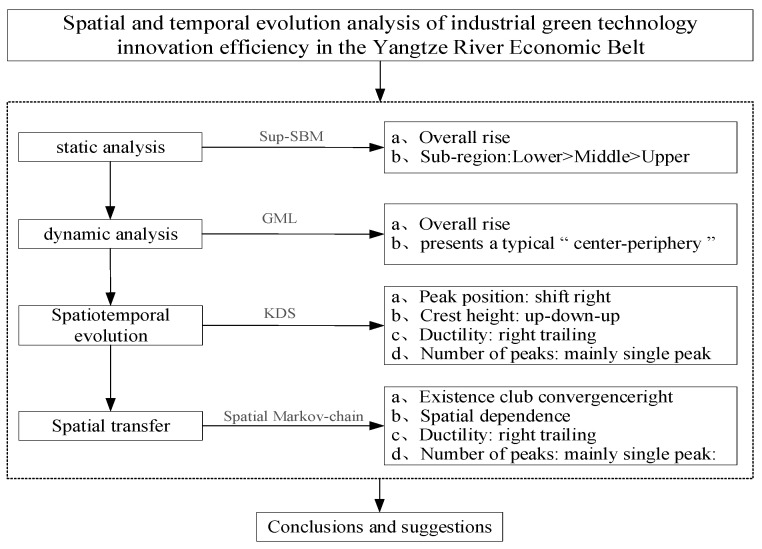
The study framework research design.

**Figure 2 ijerph-19-06361-f002:**
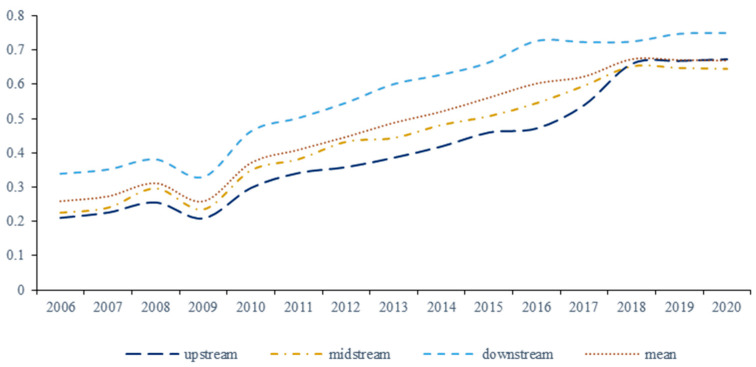
The change trend of industrial green technology innovation efficiency in the Yangtze River Economic Belt.

**Figure 3 ijerph-19-06361-f003:**
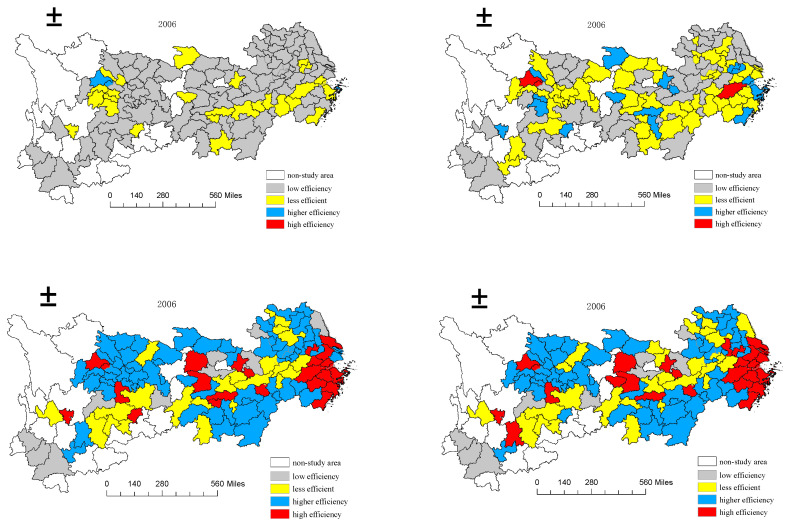
Temporal and spatial evolution of industrial green technology innovation efficiency in 110 cities in the Yangtze River Economic Belt.

**Figure 4 ijerph-19-06361-f004:**
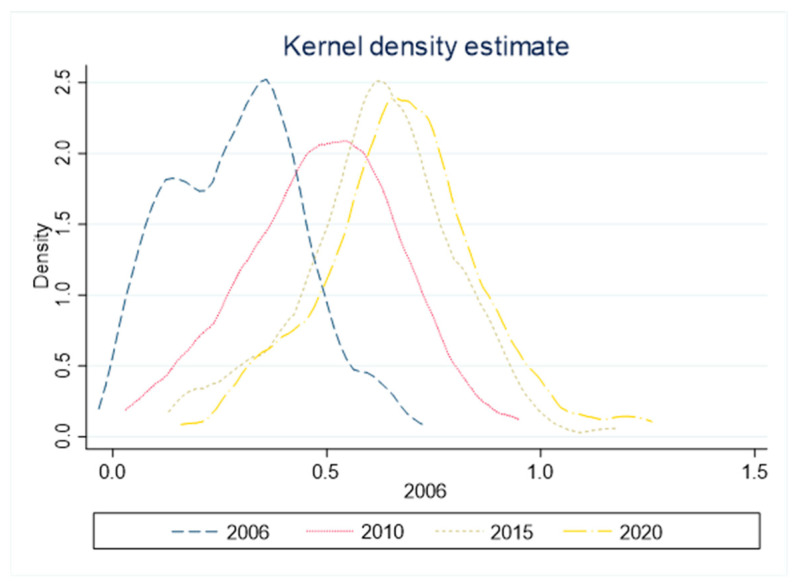
The core density distribution of the innovation efficiency of green technology in the economy and industry of the Yangtze River.

**Table 1 ijerph-19-06361-t001:** The evaluation index system of industrial green technology innovation efficiency in the Yangtze River Economic Belt.

First-Level Indicator	Secondary Indicators	Indicator Description	Unit
Input variable	R&D staff input	Full-time equivalent of R&D personnel in industrial enterprises above designated size	People/year
	R&D spending	Internal expenditure of R&D funds of industrial enterprises above a designated size	CNY 10,000
	Industrial energy input	Total industrial energy consumption	10,000 tons of standard coal
Expected output	The number of patent applications applied for	The number of effective invention patents of industrial enterprises above the designated size	Item
	The number of new product development projects	The number of new product projects of industrial enterprises above the designated size	Item
	New product sales revenue	New product sale revenues of industrial enterprises above designated size	CNY 10,000
Undesired output	Industrial waste	Industrial wastewater discharge	10,000 tons
	industrial waste	Industrial SO_2_ emissions	Billion cubic meters
	Industrial solid waste	The amount of industrial solid waste generated	10,000 tons

**Table 2 ijerph-19-06361-t002:** GML index and its decomposition of industrial green technology innovation efficiency in the Yangtze River Economic Belt.

Year	Mean			Upstream			Midstream			Downstream		
	GML	TC	EC	GML	TC	EC	GML	TC	EC	GML	TC	EC
2006–2007	1.028	1.021	1.007	1.029	1.016	1.012	1.017	1.022	0.995	1.037	1.025	1.012
2007–2008	0.852	1.069	0.797	0.838	1.028	0.815	0.828	1.076	0.769	0.889	1.102	0.806
2008–2009	0.948	1.119	0.848	0.935	1.059	0.882	0.934	1.120	0.834	0.975	1.179	0.827
2009–2010	1.221	1.121	1.088	1.235	1.108	1.114	1.182	1.127	1.049	1.246	1.130	1.102
2010–2011	1.088	1.050	1.036	1.077	1.014	1.063	1.028	1.049	0.981	1.159	1.086	1.066
2011–2012	1.108	1.127	0.984	1.171	1.097	1.067	1.081	1.135	0.951	1.074	1.150	0.934
2012–2013	1.035	1.024	1.011	1.057	1.031	1.024	1.007	1.017	0.990	1.041	1.022	1.017
2013–2014	1.010	1.066	0.947	0.971	1.016	0.955	0.991	0.922	1.075	1.067	1.104	0.965
2014–2015	1.188	1.136	1.046	1.183	1.103	1.072	1.167	1.137	1.026	1.213	1.167	1.040
2015–2016	0.901	0.845	1.066	0.913	0.826	1.104	0.878	0.848	1.035	0.911	0.860	1.059
2016–2017	1.170	1.142	1.024	1.109	1.116	0.992	1.274	1.163	1.095	1.127	1.144	0.985
2017–2018	1.095	1.091	1.005	1.049	1.038	1.010	1.109	1.108	1.001	1.127	1.125	1.001
2018–2019	1.020	1.082	0.944	1.004	1.020	0.983	1.010	1.097	0.921	1.045	1.126	0.927
2019–2020	1.093	1.104	0.991	1.024	1.021	1.003	1.107	1.127	0.982	1.146	1.163	0.986
mean	1.054	1.071	0.985	1.042	1.035	1.007	1.044	1.079	0.968	1.075	1.099	0.981

Table data source: calculated using Matlab 2014a.

**Table 3 ijerph-19-06361-t003:** Gini coefficient and decomposition results of industrial green technological innovation efficiency in the Yangtze River Economic Belt.

Year	Overall Coefficient	Regional Negini Coefficient			Gini Coefficient between Regions			Contribution Rate (%)		
		Uptream	Midstream	Downstream	Up-Midstream	Mid-Downstream	Up-Midstream	Within the Area	Interregional	Hypervariable Density
2006	0.484	0.319	0.346	0.577	0.583	0.552	0.344	31.00	41.35	27.65
2007	0.436	0.316	0.314	0.528	0.186	0.122	0.238	31.24	37.84	30.92
2008	0.427	0.313	0.311	0.493	0.302	0.184	0.252	30.42	36.01	33.57
2009	0.416	0.312	0.304	0.472	0.327	0.215	0.273	31.26	36.75	31.99
2010	0.405	0.312	0.294	0.450	0.471	0.431	0.310	32.65	35.12	32.23
2011	0.361	0.267	0.270	0.387	0.433	0.392	0.286	30.34	38.96	30.70
2012	0.324	0.258	0.236	0.354	0.394	0.351	0.256	32.24	36.71	31.05
2013	0.572	0.491	0.465	0.569	0.679	0.610	0.508	31.15	33.32	35.53
2014	0.324	0.297	0.223	0.350	0.388	0.336	0.274	31.23	34.07	34.70
2015	0.313	0.296	0.217	0.331	0.379	0.328	0.272	31.63	32.56	35.79
2016	0.308	0.294	0.220	0.329	0.372	0.322	0.276	32.53	33.02	34.45
2017	0.302	0.295	0.214	0.332	0.365	0.319	0.277	30.68	31.74	37.58
2018	0.304	0.290	0.212	0.328	0.360	0.312	0.279	32.14	30.22	36.65
2019	0.437	0.469	0.306	0.440	0.528	0.413	0.427	32.50	22.75	44.75
2020	0.353	0.348	0.254	0.360	0.436	0.341	0.323	30.38	31.36	38.26

Table data source: calculated using Matlab 2014a.

**Table 4 ijerph-19-06361-t004:** Traditional Markov and Markov probability transition matrix of industrial green technology innovation efficiency in the Yangtze River Economic Belt.

Spatial Lag Type	*t*/*t* + 1	I	II	III	IV
No lag	I	0.806	0.181	0.013	0.000
	II	0.016	0.660	0.306	0.018
	III	0.000	0.017	0.742	0.241
	IV	0.000	0.000	0.029	0.971
I	I	0.847	0.127	0.026	0.000
	II	0.174	0.720	0.106	0.000
	III	0.000	0.164	0.705	0.131
	IV	0.000	0.000	0.084	0.916
II	I	0.749	0.251	0.000	0.000
	II	0.042	0.773	0.185	0.000
	III	0.000	0.035	0.690	0.275
	IV	0.000	0.000	0.065	0.935
III	I	0.702	0.285	0.013	0.000
	II	0.000	0.793	0.217	0.000
	III	0.000	0.000	0.763	0.237
	IV	0.000	0.000	0.031	0.969
IV	I	0.637	0.318	0.031	0.014
	II	0.000	0.648	0.352	0.000
	III	0.000	0.000	0.694	0.306
	IV	0.000	0.000	0.000	1.000

Note: No lag represents the traditional Markov chain probability transition matrix. Table data source: calculated using stata.

## Data Availability

The data can be obtained from the China Industrial Statistics Yearbook, China Energy Statistics Yearbook, the China Science and Technology Statistics Yearbook, and the statistical yearbooks and statistical bulletins of the corresponding years of various provinces and cities. The data of invention patents are from the National Patent Office, and the “three wastes” in the unexpected output are mainly from the Guotaian database.

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
