# Peer review of "Spatial and Temporal Evolution Analysis of Industrial Green Technology Innovation Efficiency in the Yangtze River Economic Belt"

_ijerph, 2022, doi:10.3390/ijerph19116361_

Round 1

Reviewer 1 Report

The introduction is not very useful. Therefore, the introduction should be extended very carefully. The introduction section should be rewritten again. The introduction should highlight the study's novelty and motivation and put some literature without any useful explanation; in fact, the introduction should be clearly stated research questions and targets first. Then answer several questions: Why is the topic important (or why do you study on it)? What are the research questions? What has been studied? What are your contributions? Why is it to propose this particular method? This study's major defect is the debate or argument is not clearly stated in the introduction session.

I would suggest the author improve your theoretical discussion and arrives at your debate or argument. In addition, the background introduction should be condensed. The literature review is not presented in a good structure, and at the end of LR, you should come out with a paragraph to conclude your discussion, in this paragraph, you can highlight the novelty of your study also, it means what the LR has done and what you want to do. The literature review must highlight the novelty and contribution of the study, but these sections, which the authors provided only are related works and not literature review. Authors must carefully revise these sections.

There are several grammatical errors in the paper. 

Reviewer 2 Report

Thank you dear author(s) of the nice research. The work has a significant impact on green technology and innovation. Despite of the importance of the research the study has major findings and weakness given below: 

  1. The abstract is very detailed and  not organised
  2. The motivation of research is very scattered while research questions and goals are incomplete. 
  3. The literature review of the study is completely missing that reduced the research gap and framework. 
  4. What is the research framework?
  5. I didn't notice any theoretical consideration in the whole study. 
  6. Research methodology, specially analysing techniques and methods are well established but overall methodology is very weak and misleading. There is no appropriate analysis regarding the solution of the green growth. The study didn't consider many micro and macro level variables like green policy and regulation, green finance, green strategy, green competitiveness, market value and reputation of firms, gender and diversity impact. 
  7. The study has a sound and significant implication in the theory, policy and management while the study missed most of them. 
  8. There are no limitations and future directions. 

Therefore, it is necessary to update the study based on the above issues. Moreover, it is better to propose hypotheses for empirical testing of the data and growth along with a forecasting model. 

Reviewer 3 Report

Dear Authors,

The choice of topic is very timely and concerns the development of Green technology innovation.

After reading the article I have the following comments:

Introduction

The theoretical part is well presented. However the aim of the paper is missing, which should be at the end of this subchapter. For a better understanding of the objectives I suggest introducing research questions.

Construction of evaluation index system

Please justify more precisely the selection of indicators adopted for the study.

These indicators are different from the indicators and guidelines used in other parts of the world, e.g. Europe:

Flouros, F., V. Pistikou and V. Plakandaras. "Geopolitical risk as a determinant of renewable energy investments." Energies 15 (2022): 1498,

Commission, E. The european green deal, com(2019) 640 final, 11 december. https://ec.europa.eu/info/sites/info/files/europeangreen-deal-communication_en.pdf.: 2019,

Georgeson, L., M. Maslin and M. Poessinouw. "The global green economy: A review of concepts, definitions, measurement methodologies and their interactions." Geo: Geography and Environment 4 (2017): e00036.

Research design

Suggests introducing a scheme of research conduct

Suggests introducing the Yangtze River research area in more detail. What exactly region was included in the research and why. What are its social and economic characteristics?

Analysis of empirical results

This part of the study is very well presented and supported by the results. The discussion is sufficient, although there is no reference to foreign literature.

Aurors in the discussion refer to (306) typical " center-periphery " dual spatial structure. What is missing is a detailed description of what this means?

Conclusion

I suggest to extend the information on differences and similarities between the results of our own research and those of other researchers. What are the advantages of these studies?

Technical errors

Figures and tables do not have the source given. It should be completed.

Correct font (it is not uniform), different line breaks.

Kind regards

Round 2

Reviewer 1 Report

Authors have addressed my comments sufficiently.

Good luck!

Author Response

Comments of reviewers who have been revised and replied in the first round of review

Reviewer 2 Report

Thank you for the revised version of the research. This version improves the study a lot but still the study has limitations based on the journal quality and readers interest. 

Literature and proposition of the paper is nice but methodology is very weak specially statistical analysis. It is difficult to conclude the green technology innovation and efficiency by this approach. 

The contribution is also limited because of non directional research approach. The scientific soundness is limited. Green technology innovation and green efficiency are the matter of  policy, and technological improvement that is missed here. 
